# Sulfamethoxazole-trimethoprim plus rifampicin combination therapy for methicillin-resistant *Staphylococcus aureus* infection: An *in vitro* study

Masaki Nakamura[1,2]*, Yoshinori Tomoda[3], Masahiro Kobayashi[3], Hideaki Hanaki[2], Yuhsaku Kanoh[1]

1 Department of Laboratory Medicine, Kitasato University School of Medicine, Kitazato, Minami-ku, Sagamihara, Kanagawa 252-0374, Japan, 2 Ōmura Satoshi Memorial Institute, Kitasato University, Shirokane, Minato-ku, Tokyo 108-8641, Japan, 3 Laboratory of Clinical Pharmacokinetics, Research and Education Center for Clinical Pharmacy, Kitasato University School of Pharmacy, Shirokane, Minato-ku, Tokyo 108-8641, Japan.

* m-nakamu@kitasato-u.ac.jp

## Abstract

Methicillin-resistant *Staphylococcus aureus* (MRSA) is highly drug-resistant. The current Japanese guidelines (2019 edition) for managing and treating MRSA infections mention alternative anti-MRSA agents, including sulfamethoxazole-trimethoprim (ST) and rifampicin (RFP). Both ST and RFP are oral drugs and are expected to be effective alternatives to anti-MRSA drugs in clinical cases where anti-MRSA drugs are not indicated. Although guidelines for treating MRSA infections describe the efficacy of ST-RFP combination therapy and although it is used in clinical practice, only a limited number of *in vitro* studies have demonstrated its efficacy in pharmacokinetic/pharmacodynamic models. This study aimed to investigate the efficacy of the combination therapy of ST-RFP against MRSA in some *in vitro* models, including a pharmacokinetic/pharmacodynamic model. The MRSA strains obtained were subjected to antibiotic susceptibility tests. A checkerboard assay for drug combination synergy of ST-RFP was performed. Furthermore, a chemostat model was used to validate the combination therapy of ST-RFP as an *in vitro* pharmacokinetic/pharmacodynamic model. Viable cell counts and antibiotic concentrations were measured. The checkerboard assay showed that the ST-RFP combination had an additive effect on all strains (the lowest fractional inhibitory concentration index ranged from 0.63 to 1.00). The *in vitro* chemostat model demonstrated the usefulness of the combination of ST-RFP against MRSA, especially in ST-resistant strains. Regrowth of MRSA was observed in RFP monotherapy but not in the ST-RFP combination therapy. This study demonstrated the potential effectiveness of the ST-RFP combination against ST-resistant MRSA, providing important foundational data that may support future clinical investigations.

**Data availability statement:** All relevant data are within the paper and its Supporting Information files.

**Funding:** This study was supported by a grant from the Kitasato University School of Allied Health Sciences (Grant-in-Aid for Research Project, No. 2021-1046) to M.N. URL of the funder's website: https://www.kitasato-u.ac.jp/ahs/. This study was also supported by JSPS KAKENHI (Grant Number: JP23K09655). The funder had no role in the study design, data collection and analysis, decision to publish, or preparation of the manuscript

**Competing interests:** The authors have declared that no competing interests exist.

## Introduction

Methicillin-resistant *Staphylococcus aureus* (MRSA) is a dangerous form of drug-resistant bacteria that is the leading cause of hospital- and community-acquired infections and is associated with significant morbidity, mortality, and healthcare cost burden [1]. In Japan, six anti-MRSA agents have been approved for MRSA-associated infections, namely vancomycin (VCM), teicoplanin (TEIC), arbekacin (ABK), linezolid (LZD), daptomycin (DAP), and tedizolid (TZD) [2]. In case of difficulty or treatment failure when using any anti-MRSA agent as monotherapy, combination therapy is often considered. Some clinical trials have investigated the efficacy of combination therapies, including VCM plus β-lactams or DAP plus β-lactams [1].

Practical guidelines for the management and treatment of infections caused by MRSA (2019 Edition) [2] mention alternative anti-MRSA agents containing sulfamethoxazole-trimethoprim (ST), rifampicin (RFP), minocycline (MINO), and clindamycin (CLDM). RFP monotherapy usually leads to the development of antibiotic resistance; therefore, combination therapy is recommended when using RFP. The Japanese guidelines for the treatment of MRSA infections mention the use of combination therapy with ST and RFP to prevent antibiotic resistance. In addition, European guidelines for pediatric osteoarticular infections also describe ST plus RFP treatment for *Staphylococcus aureus* [3], and the 2011 guidelines for the treatment of MRSA infections issued by the IDSA in the United States, ST + RFP is listed as an oral treatment option for osteoarticular infections [4]. In contrast, in the 2021 MRSA treatment guidelines in the United Kingdom, the combination of RFP with ST was once recommended for osteoarticular infections involving biofilms; however, the guidelines only state that there is no clear evidence for this [5]. Recently, some reports have described the efficacy of ST plus RFP combination therapy. Unlike VCM and DAP, which are administered intravenously, ST and RFP are administered orally. Oral treatment with ST plus RFP has been administered to asymptomatic carriers of MRSA [6,7]. David et al. [8] and Daniela et al. [9] reported that ST plus RFP combination therapy is, to some extent, effective in clearing MRSA in patients with cystic fibrosis. Anna-Karin et al. reported that the combination of ST and RFP was more effective in eliminating pharyngeal MRSA carriage than topical treatment with mupirocin alone [10].

There is no consensus on the efficacy of ST and RFP combination therapy for the treatment of invasive MRSA infections, such as infective endocarditis or bacteremia. There are a few reports of MRSA infective endocarditis with lung abscesses treated with VCM plus ST and RFP [11]. The adjunctive rifampicin for *Staphylococcus aureus* bacteremia (ARREST) study, a multicenter, randomized, double-blind, placebo-controlled trial, assessed the efficacy of adjunctive RFP [12]. The ARREST study revealed no added advantages of adjunctive RFP; however, various anti-MRSA agents were included in the study backbone antibiotic therapies apart from ST. Hence, the efficacy of ST plus RFP combination therapy alone against MRSA bacteremia remains unclear. Therefore, basic studies on ST plus RFP combinations, such as checkerboard assays or *in vitro* pharmacodynamic models, are valuable for supporting the optimum clinical use of this therapy.

*In vitro* pharmacodynamic models can simulate *in vivo* human pharmacokinetic profiles and more accurately assess the efficacy of antibiotics, including the emergence of drug resistance [13,14]. These models have also been used to study MRSA infections [15–18]. Reports on the benefits of ST plus RFP combination therapy have increased in recent years. However, most of these are clinical studies, and only a few basic research studies are available. In this study, we investigated the efficacy of ST plus RFP against MRSA strains using a checkerboard assay and an *in vitro* chemostat model based on basic laboratory research.

## Materials and methods

### Bacterial strains and antibiotics

Methicillin-susceptible *S. aureus* (MSSA) ATCC 29213 and MRSA ATCC 43300 were the two reference strains obtained from the American Type Culture Collection (ATCC, Manassas, Virginia, USA). Two ST-resistant (ST-R) strains, KAM444 and KAM636, were provided by the Research Center for Infection Control, Omura Satoshi Memorial Institute (Minato-ku, Tokyo, Japan). One low-ST-susceptible (ST-LS) strain KAM 1160 was supplied by Kitasato University Hospital (Sagamihara, Kanagawa, Japan).

Sulfamethoxazole (SMX), trimethoprim (TMP), and RFP were purchased from Wako Chemical Co., Ltd. (Osaka, Japan). A stock solution of each antibiotic for the checkerboard and pharmacodynamic assays was prepared using a dimethyl sulfoxide (DMSO; Wako Chemical Co., Ltd.) solution. The ST solution was prepared using SMX and TMP at a mixing ratio of 19:1 (w/w). To stabilize RFP, ascorbic acid (Wako Chemical Co., Ltd.) was added to a calcium-adjusted Mueller-Hinton broth (CAMHB; Becton, Dickinson and Company, NJ, USA) at a final concentration of 0.1 mg/mL for the pharmacodynamic assay.

### Antibiotic susceptibility tests

The antibiotic susceptibility of each strain was tested using ready-made dry plates, DP42 (Eiken Chemical Co., Ltd., Tokyo, Japan). The plates contained 21 antimicrobial agents, whose minimum inhibitory concentration (MIC) assays were measured for the 12 drugs as follows: oxacillin (MPIPC), cefazolin (CEZ), cefoxitin (CFX), erythromycin (EM), CLDM, MINO, ST, RFP, VCM, TEIC, LZD, and DAP. A bacterial suspension containing $5 \times 10^4$ CFU of each strain was inoculated into each well and incubated at 35 °C in room air for 18 h according to the guidelines of the Clinical and Laboratory Standards Institute (CLSI) M100 [19]. Three replicates of MIC assays were performed for each chemical entity. The MICs of ST and RFP, the key drugs in this study, were also measured using the E-test (Sysmex bioMérieux, France).

### Checkerboard assay

Checkerboard assays for drug combination synergy studies were performed using 96-well plates as previously described [20]. Antibiotics were serially diluted in CAMHB along the abscissa for ST or ordinate for RFP. The final concentration of each antibiotic was as follows: ST concentration, 0.031–2 µg/mL for the reference strains and 0.125–8 µg/mL for the ST-R and ST-LS strains; RFP concentration, 0.0005–0.032 µg/mL for all strains. The ST concentrations represent the TMP concentrations. A bacterial suspension containing $5 \times 10^4$ CFU of each strain was inoculated into each well and incubated at 35 °C for 18 h.

The fractional inhibitory concentration (FIC) index was calculated using the MIC of the antibiotic alone and in combination as follows: FIC index = [MIC$_{\text{(ST in combination)}}$/ MIC$_{\text{(ST alone)}}$] + [MIC$_{\text{(RFP in combination)}}$/ MIC$_{\text{(RFP alone)}}$]. The efficacy of the synergy effect was assessed according to the score of the FIC index as follows: an FIC index of <0.5 indicated synergism, ≥0.5 to ≤1 indicated additive effects, >1 to <2 indicated indifference, and ≥2 indicated antagonism [21]. Checkerboard assays were performed in triplicate for each combination.

### *In vitro* pharmacodynamic model

A one-compartment *in vitro* chemostat model was used to validate combination with ST-RFP. PASS-400 (DAI NIPPON SEIKI Co., Ltd., Kyoto, Japan) was used as a chemostat. Each strain was inoculated into a glass reactor containing 200 mL of CAMHB. The reactors were incubated at 37 °C in a water bath with constant stirring. The initial bacterial count was $10^6$ CFU/mL. Antibiotics and CAMHB were infused into the reactors via antibiotic pumps to achieve pharmacokinetic drug concentrations. Pharmacokinetic parameters were set based on the standard administration data for healthy individuals, according to the pharmaceutical interview forms for each drug (ST: https://www.info.pmda.go.jp/go/interview/2/343018_6290100D1088_2_023_1F.pdf/; RFP: https://pins.japic.or.jp/pdf/medical_interview/IF00000369.pdf), which are listed in Table 1. Samples were obtained from each reactor at time points of 0, 2, 4, 8, 12, and 24 h to measure the viable cell count and drug concentration. This treatment was performed in duplicate to ensure reproducibility.

### Measurement of viable cell count

To reduce the concentration of antibiotics in the samples, the harvested samples were immediately diluted with normal saline. Serially diluted aliquots were then inoculated in trypticase soy agar and incubated at 37 °C for 24 h. Following incubation, the number of colonies was counted, and viable cell counts were calculated. The lower limit of detection was 10 CFU/mL. For the surviving colonies, MICs of ST and RFP were determined using the E-test to assess the development of resistance.

### Measurement of sulfamethoxazole, trimethoprim, and rifampicin concentrations

SMX, TMP, and RFP concentrations were measured using reverse-phase high-performance liquid chromatography (HPLC). The concentration of each drug in the KAM636 samples was studied at 0, 2, 4, 8, 12, and 24 h, and that in the samples of the other strains were measured at 2, 4, and 8 h. Standard RFP solutions were prepared at concentrations of 0.1, 0.2, 0.5, 1, 5, and 10 µg/mL by diluting with CAMHB. Standard solutions of ST were prepared at concentrations of 10/0.5, 25/1.25, and 50/2.5 µg/mL by dilution with CAMHB. The HPLC system consisted of an LC-20AD pump, DGU-20A degasser, SIL-20A autosampler, CTO-20AC column oven, SPD-M20A detector, and LabSolutions® ver. 5.81 for data analysis (Shimadzu Corporation, Kyoto, Japan). Separation was performed with Inertsil C8-3 (4.6 mm i.d. × 150 mm) (GL Science Inc., Tokyo, Japan) at 30 °C. For RFP analysis, a mobile phase consisting of 10 mM phosphate buffer (pH 7.2): acetonitrile in a 60:40 (v/v) ratio was used at a flow rate of 1.2 mL/min, and the absorbance was measured at 340 nm. Further, for SMX and TMP analysis, a mobile phase consisting of 50 mM acetate buffer (pH 5.7): methanol in a 73:27 (v/v) ratio was used at a flow rate of 1.2 mL/min, and the absorbance was measured at 240 nm.

## Results

### Antibiotic susceptibility results

The results of the antibiotic susceptibility tests are presented in Table 2. The MIC of the antibiotics used for quality control (QC) organisms, MSSA ATCC 29213, was in the range of the QC MIC for all antibiotics except for S/A and ABK, which are

**Table 1. Pharmacokinetic parameters in the chemostat model.**

| Antibiotics | Dose of administration | Setpoints of the pharmacokinetic parameters | | | |
|---|---|---|---|---|---|
| | | $C_{max}$ (µg/mL) | $T_{max}$ (hour) | $T_{1/2}$ (hour) | Interval (hour) |
| ST | 160 mg q12h | 2.5* | 3.4 | 7.5 | 12 |
| RFP | 450 mg q24h | 8.0 | 1.9 | 2.2 | 24 |

ST, Sulfamethoxazole/Trimethoprim; RFP, Rifampicin

$C_{max}$, maximum concentration; $T_{max}$, time of maximum concentration; $T_{1/2}$, half-life

*concentration of ST was shown as trimethoprim

**Table 2. Antimicrobial susceptibility of strains in this study.**

| Strain | MIC (µg/mL) | | | | | | | | | | | |
|---|---|---|---|---|---|---|---|---|---|---|---|---|
| | MPIPC | CEZ | CFX | EM | CLDM | MINO | ST* | RFP | VCM | TEIC | LZD | DAP |
| Reference strain | | | | | | | | | | | | |
| MSSA ATCC 29213(QC) | ≤0.25 | ≤2 | ≤4 | ≤0.25 | ≤0.25 | ≤1 | ≤0.5 | ≤0.5 | ≤0.5 | ≤0.5 | 2 | ≤0.25 |
| | | | | | | | (0.047) | (0.012) | | | | |
| MRSA ATCC 43300 | >4 | 16 | >8 | >4 | >2 | ≤1 | ≤0.5 | ≤0.5 | 1 | ≤0.5 | 2 | ≤0.25 |
| | | | | | | | (0.047) | (0.006) | | | | |
| ST-R clinical isolate | | | | | | | | | | | | |
| KAM444 | >4 | >16 | >8 | >4 | >2 | 4 | >2 | ≤0.5 | 1 | ≤0.5 | 1 | 0.5 |
| | | | | | | | (>32) | (0.008) | | | | |
| KAM636 | >4 | ≤2 | >8 | 0.5 | ≤0.25 | ≤1 | >2 | ≤0.5 | 1 | ≤0.5 | 2 | 0.5 |
| | | | | | | | (3) | (0.012) | | | | |
| KAM1160 | >4 | 16 | >8 | ≤0.25 | ≤0.25 | ≤1 | >2 | ≤0.5 | 1 | ≤0.5 | 4 | 1 |
| | | | | | | | (3) | (0.006) | | | | |

MRSA, Methicillin-resistant *Staphylococcus aureus*; MSSA, Methicillin-susceptible *Staphylococcus aureus*; MIC, minimum inhibitory concentration; QC, quality control; MPIPC, oxacillin; CEZ, cefazolin; CFX, cefoxitin; EM, erythromycin; CLDM, clindamycin; MINO, minocycline; ST, sulfamethoxazole-trimethoprim; ST-R, ST-resistant; RFP, rifampicin; VCM, vancomycin; TEIC, teicoplanin; LZD, linezolid; DAP, daptomycin.

The MICs of ST and RFP in brackets indicate values measured by E-test

*concentrations of ST were shown as trimethoprim

not described in the CLSI document M100 ED32 [19]. The reference strain of MRSA, ATCC 43300, was resistant to both MPIPC and CFX, indicating methicillin resistance. Both ST-R clinical isolates were resistant to ST and the ST-LS strain was intermediate resistance in the E-test. All clinical isolates were susceptible to RFP.

## Combined effect of ST and RFP

Bacterial growth on the checkerboard is shown in S1 Fig. The reference strains, ATCC 29213 and ATCC 43300, showed an MIC of 0.125 µg/mL for ST and 0.008 µg/mL for the single RFP drugs. In contrast, both strains showed lower ST MIC (0.063 µg/mL) and RFP MIC (0.004 or 0.002 µg/mL) with the combination of ST and RFP. In all ST-R strains, the ST MIC was >8 µg/mL, and the RFP MIC was 0.016 or 0.008 µg/mL for a single drug. In contrast, both ST MIC and RFP MIC were reduced to 8 or 2 µg/mL and 0.008 or 0.004 µg/mL, respectively, in the combination. In ST-LS strain, the ST MIC was 2 µg/mL, and the RFP MIC was 0.008 µg/mL for a single drug. In contrast, both ST MIC and RFP MIC were reduced to 1 µg/mL and 0.004 µg/mL, respectively, in the combination. Table 3 presents the lowest FIC indices and interpretations. The

**Table 3. Interaction of sulfamethoxazole/trimethoprim and rifampicin in the checkerboard assay.**

| Strain | MIC of drugs alone (µg/mL) | | MIC of drugs in combination (µg/mL) | | | |
|---|---|---|---|---|---|---|
| | ST* | RFP | ST* | RFP | Lowest FIC index | Interpretation |
| MSSA ATCC29213 | 0.125 | 0.008 | 0.063 | 0.004 | 1.00 | ADD |
| MRSA ATCC43300 | 0.125 | 0.008 | 0.063 | 0.002 | 0.75 | ADD |
| KAM444 | >8 | 0.008 | 8 | 0.004 | 1.00 | ADD |
| KAM636 | >8 | 0.016 | 2 | 0.008 | 0.63 | ADD |
| KAM1160 | 2 | 0.008 | 1 | 0.004 | 1.00 | ADD |

MRSA, Methicillin-resistant *Staphylococcus aureus*; MSSA, Methicillin-susceptible *Staphylococcus aureus*; MIC, minimum inhibitory concentration; FIC, fractional inhibitory concentration; ST, Sulfamethoxazole/Trimethoprim; RFP, Rifampicin; ADD, additive effect;

*concentrations of ST were shown as trimethoprim

lowest FIC index of all strains was within the range of 0.5 to 1.0, which indicated an additive effect following the combination of ST and RFP.

### Combination therapy of ST and RFP using the *in vitro* pharmacodynamic model

The results of the ST and RFP combination therapy in the pharmacodynamic model are shown in Fig 1. In all strains, combination therapy with ST and RFP caused more than 3-log$_{10}$ kills from the initial inoculum within 24 h. Single ST therapy showed the same reduction as ST plus RFP combination therapy in ST-susceptible reference strains (Fig 1A, B); however, it was ineffective in the ST-R and ST-LS strains (Fig 1C-E). Furthermore, a single therapy with RFP was effective in all strains until 12 h; however, regrowth was observed at 24 h.

### Induction of rifampicin resistance by *in vitro* chemostat experiments

The MICs of the strains that showed regrowth during the *in vitro* chemostat experiments were measured using the E-test. Table 4 shows the MICs of each strain. Only the regrown strains treated with RFP showed an increase in RFP MIC.

### Concentrations of sulfamethoxazole, trimethoprim, and rifampicin

The actual concentrations of SMX, TMP, and RFP in the KAM636 samples in the combination condition at 2 h, close to T$_{max}$, were 40.11, 2.45, and 7.94 µg/mL, respectively. These concentrations ranged from 93–107% of the simulated concentrations (Fig 1F). The combined drug concentrations of the samples (ST and RFP) were similar to those of ST or RFP alone. Furthermore, the actual concentrations in the samples were similar to the simulated concentrations at other times (S1 Table). In evaluations similar to those of the other strains used in this study, the actual SMX, TMP, and RFP concentrations at 2, 4, and 8 h were similar (S2 Table).

## Discussion

The practical guidelines for the management and treatment of infections caused by MRSA (2019 Edition) describe the combination therapy of ST plus RFP, and some clinical studies have reported the use of this combination therapy [8–10]. However, only a few reports have focused on basic laboratory research on this treatment modality, and its effectiveness remains controversial [22]. This study investigated the combined effects of ST and RFP on MRSA strains, including ST-R strains. The checkerboard assay revealed the additive effect of ST and RFP against all strains, and the *in vitro* pharmacodynamic model showed the effectiveness of the ST plus RFP combination therapy, particularly against ST-R and ST-LS MRSA.

Regarding the *S. aureus* strains used in the checkerboard assay, the MICs of ST and RFP in combination were lower than those of monotherapies against all strains. In particular, ST-R strains could not be inhibited by ST alone, whereas the combination of ST and RFP inhibited the growth of ST-R strains. Therefore, we inferred that the ST plus RFP combination therapy might be effective against ST-R MRSA strains. Although a previous study reported that LVX + RFP was superior to ST + RFP *in vitro* [18], this study evaluated the effect of ST + RFP on biofilms, which is different from our study. The additive effect of ST and RFP shown in this study is a new finding regarding the combination therapy with ST and RFP *in vitro*.

A report by the Japan Nosocomial Infections Surveillance (JANIS) revealed the frequency of resistance to anti-MRSA agents, and 0.6% of the MRSA strains were ST-R strains in 2021 [23]. The pharmacodynamic model in this study showed the usefulness of the combination therapy of ST plus RFP against MRSA, especially the ST-R strains. The Japanese guidelines for treating MRSA infections state that the most effective setpoints of pharmacokinetics/pharmacodynamic profiles are unclear for ST or RFP [2]. The parameters of the pharmacokinetic/pharmacodynamic profiles in this study were the usual dosages of oral ST and RFP in healthy individuals. Therefore, this *in vitro* study suggests that ST-R MRSA strains may respond to the ST plus RFP combination at standard concentrations, indicating its potential utility for future clinical evaluation.

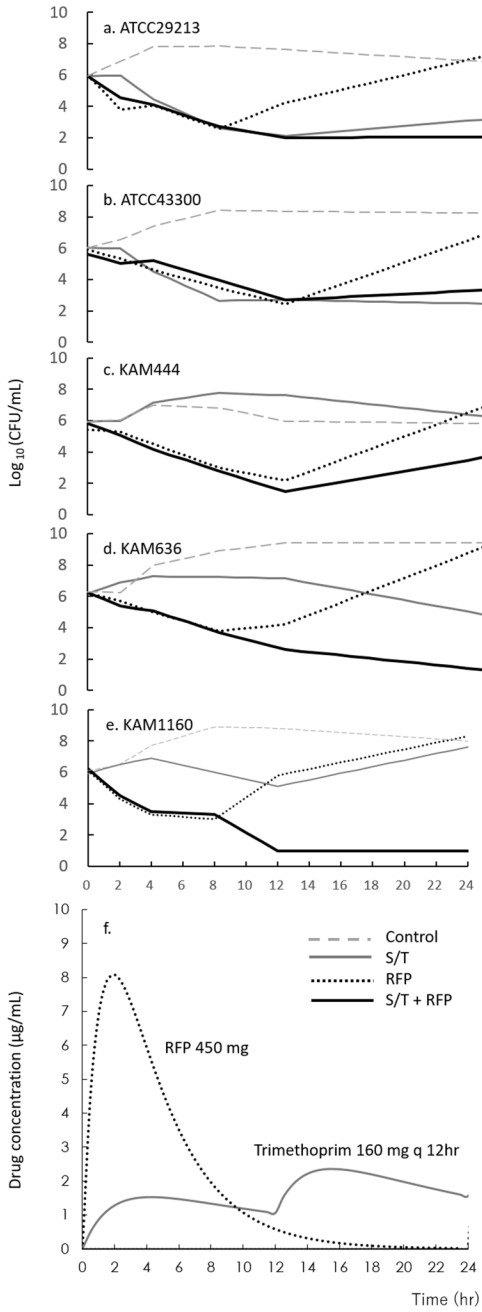

**Fig 1. *In vitro* chemostat experiments of ST plus RFP against ST-R MRSA isolates.** ATCC29213 (A) and ATCC43300 (B) are reference strains of MSSA and MRSA, respectively. KAM444 (C) and KAM636 (D) are ST-R, and KAM1160 (E) is ST-LS MRSA isolates. The drug concentration (F) indicates the reconstruction curve from the measured data. The concentration of ST was expressed as the concentration of trimethoprim. MRSA, Methicillin-resistant *Staphylococcus aureus*; MSSA, Methicillin-susceptible *Staphylococcus aureus* ST, Sulfamethoxazole-Trimethoprim; RFP, Rifampicin; ST-resistant, ST-R.

In this study, RFP monotherapy was effective in the early stages; however, bacterial regrowth occurred after 12 h of treatment. Typically, monotherapy with RFP is not recommended because it induces drug resistance. In this study, we confirmed that the MIC value of RFP clearly increased in the strains after monotherapy with RFP. The addition of ST

**Table 4. MICs of surviving colonies after the *in vitro* pharmacodynamic model.**

| strain | Treated with RFP (µg/mL) | | Treated without drug (µg/mL) | |
|---|---|---|---|---|
| | ST | RFP | ST | RFP |
| KAM444 | >32 | >32 | >32 | 0.006 |
| KAM636 | 4 | >32 | 3 | 0.016 |
| KAM1160 | 3 | >32 | 4 | 0.012 |

ST, Sulfamethoxazole/Trimethoprim; RFP, Rifampicin

to RFP inhibited the regrowth of all strains. A previous report showed that the combination of RFP and TMP effectively prevented the development of RFP resistance in a murine *S. aureus* infection model [24]. Moreover, a review of clinical studies reported that oral RFP is an effective agent for preventing RFP resistance when combined with ST or MINO [25]. The *in vitro* pharmacodynamic model used in this study showed a preventive effect against RFP resistance, which is consistent with the results of previous reports. For the KAM636 and KAM1160 strains, ST monotherapy was somewhat effective, indicating that ST had a lower MIC against the KAM636 and KAM1160 strains than against the KAM444 strain (Table 2).

Combination therapy with ST and RFP is becoming increasingly imperative for MRSA infections. Antibiotic exposure induces persistent intracellular *S. aureus* infection, contributing to therapeutic failure [26]. Moreover, ST plus RFP combination therapy was more effective at exterminating intracellular MRSA than VCM monotherapy [27].

In Japan, LZD and TZD are the only anti-MRSA agents used in oral medicine. Even when there is no indication of MRSA infection, ST and RFP are considered alternative anti-MRSA agents. An early switch from intravenous to oral antimicrobials has various benefits, including reducing the risk of catheter-related infections and the cost of antimicrobial therapy without compromising clinical outcomes [28]. Some reports regarding clinical trials of ST plus RFP versus LZD alone for treating MRSA infection showed that ST-RFP combination therapy is not inferior to LZD monotherapy [29]. In addition, ST plus RFP combination therapy is more cost-effective than LZD in treating MRSA infections [30].

This study had some limitations. The results of the experiments in this study were evaluated for only up to 24 h, and only two reference and three clinically isolated strains were assessed in this study. Therefore, the time taken for the ST-RFP combination therapy to inhibit the development of resistant strains is unclear. More strains and longer chemostat model experiments are required. Furthermore, the pharmacodynamic model in this study represents the antibiotic concentration curve for blood, not the tissues. Thus, the results could be extrapolated to sepsis or bacteremia but not to other infectious diseases. The advantage of ST-RFP combination therapy was observed in ST-R and ST-LS MRSA isolates but not in the reference MRSA strain. Moreover, RFP-R and ST-RFP double-resistant isolates were not evaluated in this study. Therefore, cases that need ST-RFP combination therapy may not be so common. However, we would like to emphasize that this study was based solely on *in vitro* experiments and was not intended to directly suggest changes to the current clinical treatment strategies. Nonetheless, our findings suggest potential clinical applicability in specific scenarios, such as when the causative organism cannot be isolated or its MIC to ST is unknown, and oral therapy is preferred. In such cases, ST-RFP combination therapy may serve as a viable treatment option, although further clinical validation is required.

## Conclusions

In this study, we demonstrated the effectiveness of ST plus RFP combination against MRSA using a checkerboard assay analysis and an *in vitro* pharmacodynamic model. Studies on ST plus RFP therapy are insufficient, and its effectiveness is controversial. Therefore, the results of this study are worthwhile from the viewpoint of accumulating further evidence, although they cannot be directly extrapolated to clinical settings.

## Supporting information

**S1 Fig. Checkerboard assays for sulfamethoxazole-trimethoprim and rifampicin combination therapy.** The gray wells indicate bacterial growth, and the blank wells indicate the inhibition of bacterial growth. ATCC29213 (a) and ATCC43300 (b) are reference strains of MSSA and MRSA, respectively. KAM444 (c) and KAM636 (d) are ST-R, and KAM1160 (e) is ST-LS MRSA isolates.
(TIF)

**S1 Table. Simulator and actual concentrations (µg/mL) of sulfamethoxazole, trimethoprim and rifampicin.** The combination represents sulfamethoxazole-trimethoprim and rifampicin therapy, and alone represents sulfamethoxazole-trimethoprim or rifampicin therapy.
(DOCX)

**S2 Table. Actual concentrations (µg/mL) of sulfamethoxazole, trimethoprim and rifampicin in each strain study.**
(DOCX)

## Acknowledgments

The authors thank the staff of the Kitasato University School of Allied Health Sciences: Hidero Kitasato, Shotaro Maehana, Ryotaro Eda, and Makoto Kubo. The authors also thank Hidehito Matsui, Ryo Nakayama, and Yumiko Suzuki for providing bacterial strains and technical assistance with the chemostat model.

## Author contributions

**Conceptualization:** Masaki Nakamura.

**Data curation:** Masaki Nakamura.

**Formal analysis:** Masaki Nakamura, Yoshinori Tomoda, Masahiro Kobayashi.

**Investigation:** Masaki Nakamura, Yoshinori Tomoda, Masahiro Kobayashi.

**Supervision:** Hideaki Hanaki, Yuhsaku Kanoh.

**Visualization:** Masaki Nakamura, Yoshinori Tomoda, Masahiro Kobayashi.

**Writing – original draft:** Masaki Nakamura.

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
