## [Decision Letter · Decision Letter 0]

16 Oct 2024

PONE-D-24-38956Sulfamethoxazole-trimethoprim plus rifampicin combination therapy for MRSA infection: An in vitro studyPLOS ONE

Dear Dr. Nakamura,

Thank you for submitting your manuscript to PLOS ONE. After careful consideration, we feel that it has merit but does not fully meet PLOS ONE’s publication criteria as it currently stands. Therefore, we invite you to submit a revised version of the manuscript that addresses the points raised during the review process.

We look forward to receiving your revised manuscript.

Kind regards,

Abdelwahab Omri, Pharm B, Ph.D, Laurentian University, Canada

Academic Editor

PLOS ONE

Journal Requirements:

Additional Editor Comments :

Dear Authors,

Thank you for submitting your manuscript for consideration. We have now received comments from three reviewers, and based on their feedback, we invite you to submit a major revision of your work.

The reviewers found your study on the efficacy of a combination of SXT and RIF against MRSA to be interesting and potentially valuable. However, they have raised several important points that need to be addressed before the manuscript can be considered for publication.

Major Points for Revision:

1. Sample Size and Strain Selection: Expand your study to include a larger sample size, encompassing both clinical and reference strains. This will strengthen the validity of your findings.

2. Literature Review: Re-evaluate your claim regarding the lack of in vitro studies for the TMP/SMX+rifampin combination in pharmacokinetic/pharmacodynamic models. A reviewer has pointed out at least one relevant study (El Haj et al., 2018) that you should acknowledge and discuss.

3. Guidelines: Broaden your discussion of guidelines to include international perspectives, not just those from Japan. This will increase the relevance of your work for a global audience.

4. Strain Characterization: Clarify the discrepancies in MIC values for your clinical strains, particularly regarding the strain with an intermediate MIC to TMP/SMX. Discuss the implications of these differences on your results and conclusions.

5. Resistance Development: Address whether you checked for the development of rifampin resistance in your experiments, particularly for the KAM444 strain that regained growth.

6. Clinical Relevance: Provide a more nuanced discussion of the clinical applicability of your findings, particularly regarding the use of this combination therapy for TMP/SMX-resistant strains and in cases of sepsis or bacteremia

Minor Points:

1. Improve the quality of Figure 1, which was reported as blurry.

2. Address the specific language and formatting suggestions made by the reviewers, including:

a. Removing the word 'therapy' where inappropriate

b. Correcting alignment issues in Table 2

c. Reconsidering certain assertive statements

d. Using passive forms where appropriate

e. Properly referencing laboratory products

Additional Suggestions:

1. Explain why only two replicates were conducted instead of three.

2. Consider removing or revising the sentence about a previous report (lines 260-261) as suggested by Reviewer 1.

3. Add the web address mentioned on line 267 to your reference list.

Please provide a point-by-point response to the reviewers' comments along with your revised manuscript. We look forward to receiving your revised submission.

Sincerely,

Reviewers' comments:

Reviewer's Responses to Questions

**Comments to the Author**

1. Is the manuscript technically sound, and do the data support the conclusions?

Reviewer #1: Yes

Reviewer #2: Partly

Reviewer #3: Partly

2. Has the statistical analysis been performed appropriately and rigorously? 

Reviewer #1: N/A

Reviewer #2: Yes

Reviewer #3: N/A

3. Have the authors made all data underlying the findings in their manuscript fully available?

Reviewer #1: Yes

Reviewer #2: Yes

Reviewer #3: Yes

4. Is the manuscript presented in an intelligible fashion and written in standard English?

Reviewer #1: Yes

Reviewer #2: Yes

Reviewer #3: Yes

5. Review Comments to the Author

Reviewer #1: Dear authors,

The study is generally well-planned and the manuscrpit is generally well-written. I have few recommendations about the manuscript:

Line 44 and 312: Remove the word ‘therapy’ because this is not a clinical study.

Line 47: Write ‘against’ instead of ‘therapy of’.

Table 2: Some alignment problems exist in some columns of Table 2. These should be corrected.

Line 260-261: I recommend the authors to remove the sentence ‘‘In a previous report, RFP….’’ and ‘‘reference 19’’. In the cited report, although rifampicin and minocycline considerably decreased both WT and SCVs, both bacterial counts recovered to an initial number 48 hours later.

Line 267: A number can be given for the web adress and this reference can be added to references list.

Line 273-274: This sentence is a bit assertive. This was an in vitro study and the number of strains tested in this study were low. You can use a more approprite sentence or you should support this sentence with information obtained from clinical studies.

Reviewer #2: The article by Masaki Nakamura et al. presents the findings of an in vitro study investigating the efficacy of a combination of SXT and RIF against MRSA.

The article presents some intriguing findings, yet it is imperative to substantiate these outcomes with a more extensive sample size, encompassing both clinical and reference strains.

In the event of a resubmission, other factors will have to be taken into consideration.

It is preferable to utilise passive forms.

Please clarify why only two replicates were conducted, rather than three.

The sources of the laboratory products in question are not adequately referenced.

Reviewer #3: I carefully read the interesting manuscript describing the in vitro activity of TMP/SMX and rifampin combination for MRSA in 4 bacterial strains. There are limited therapeutic options for MRSA and sometimes even susceptible strains may persist antibiotic therapy. Therefore, combination therapy is often used. The methods are sound and overall, the manuscript is well written. There are, however, some issues to address.

Major comments:

1. The authors claim there is no in vitro studies for the TMP/SMX+rifampin combination in pharmacokinetic/pharmacodynamic models. That is not accurate. El Haj et al showed enhanced anti-biofilm capacity but raised concerns that rifampin resistance emerges. Combination of linezolid and rifampin protected against resistance (Cristina El Haj, Oscar Murillo, Alba Ribera, Nuria Lloberas, Joan Gómez-Junyent, Fe Tubau, Pere Fontova, Carme Cabellos, Javier Ariza, Evaluation of linezolid or trimethoprim/sulfamethoxazole in combination with rifampicin as alternative oral treatments based on an in vitro pharmacodynamic model of staphylococcal biofilm, International Journal of Antimicrobial Agents, Volume 51, Issue 6, 2018, Pages 854-861, https://doi.org/10.1016/j.ijantimicag.2018.01.014).

2. The authors refer to guidelines from Japan. It would be helpful for the broader readership to refer to other local and international guidelines. These are a few examples:

Catherine Liu, Arnold Bayer, Sara E. Cosgrove, Robert S. Daum, Scott K. Fridkin, Rachel J. Gorwitz, Sheldon L. Kaplan, Adolf W. Karchmer, Donald P. Levine, Barbara E. Murray, Michael J. Rybak, David A. Talan, Henry F. Chambers, Clinical Practice Guidelines by the Infectious Diseases Society of America for the Treatment of Methicillin-Resistant Staphylococcus aureus Infections in Adults and Children, Clinical Infectious Diseases, Volume 52, Issue 3, 1 February 2011, Pages e18–e55, https://doi.org/10.1093/cid/ciq146

Nicholas M Brown, Anna L Goodman, Carolyne Horner, Abi Jenkins, Erwin M Brown, Treatment of methicillin-resistant Staphylococcus aureus (MRSA): updated guidelines from the UK, JAC-Antimicrobial Resistance, Volume 3, Issue 1, March 2021, dlaa114, https://doi.org/10.1093/jacamr/dlaa114

3. The authors investigated two reference ATCC strains susceptible to both TMP/SMX and rifampin and two clinical strains resistance to TMP/SMX and susceptible to rifampin. These two clinical strains, however, differed in their MICs to TMP/SMX which were 3 and >32. The EUCAST and CLSI breakpoints for TMP/SMX are >4 for resistance and 2> for susceptible. Thus, one of the clinical strains tested was actually non-susceptible with intermediate MIC and the other resistant. Using the checkerboard method this intermediate strain was resistant with MIC >8 and reduced to 2 when rifampin was added. Can the authors comment son these differences?

4. The lowest fractional inhibitory concentration index for combination therapy was 0.5-1 indicating an additive value and chemostat-based kill curves showed increased efficacy of combination therapy in the resistant strains. This was not evident for the susceptible reference strains. Still, the fully resistant strain (KAM444) regained growth (but at a slower rate). Did the authors check for development of resistance for rifampin in this experiment?

5. As the authors note, TMP/SMX resistant strains are rare and TMP/SMX alone is often used as empiric therapy when MRSA is suspected. As a clinician, I wouldn’t know why I should use TMP/SMX in combination when the isolate is actually resistant to TMP/SMX. I would definitely not think of using this combination for sepsis or bacteremia as the authors suggest.

Minor comments

1. Figure 1 is blurry

6. PLOS authors have the option to publish the peer review history of their article (what does this mean? ). If published, this will include your full peer review and any attached files.

**Do you want your identity to be public for this peer review?** For information about this choice, including consent withdrawal, please see our Privacy Policy .

Reviewer #1: No

Reviewer #2: No

Reviewer #3: **Yes: ** Oren Gordon

---

## [Author Response · Author response to Decision Letter 0]

15 Apr 2025

Response to Reviewers

We sincerely thank the Editor and reviewers for their thorough evaluation of our manuscript and constructive feedback. We have revised the manuscript accordingly. Below we provide a point-by-point response to each comment. The reviewer’s comments are quoted, and our responses follow each comment.

Response to Editor

Major Point 1: Sample Size and Strain Selection

Comment (Editor): “Expand your study to include a larger sample size, encompassing both clinical and reference strains. This will strengthen the validity of your findings.”

Response: We agree with the importance of a broader sample. In the revised manuscript, we have added one additional ST-low-susceptible MRSA clinical isolate (KAM1160) to address the small sample size. We have explicitly noted in the Discussion section that the limited strain selection is a study limitation, and we emphasize the need to examine more strains in future studies.

Major Point 2: Literature Review

Comment (Editor): “Re-evaluate your claim regarding the lack of in vitro studies for the TMP/SMX + rifampin combination in pharmacokinetic/pharmacodynamic models. A reviewer has pointed out at least one relevant study (El Haj et al., 2018) that you should acknowledge and discuss.”

Response: Thank you for this suggestion. We have revised the manuscript to acknowledge the prior in vitro study by El Haj et al. (2018) that examined the trimethoprim-sulfamethoxazole (TMP/SMX) + rifampin combination in a pharmacokinetic/pharmacodynamic (PK/PD) biofilm model. In the Discussion section of the revised manuscript, we now cite this study and compare our methods and findings with those of El Haj et al.

Major Point 3: Guidelines

Comment (Editor): “Broaden your discussion of guidelines to include international perspectives, not just those from Japan. This will increase the relevance of your work for a global audience.”

Response: We have expanded our manuscript to include international guidelines as suggested. In addition to the Japanese guidelines (2019) that were originally cited, the Introduction and Discussion of the revised manuscript now reference key international guidelines on MRSA treatment. Specifically, we have added the 2011 Infectious Diseases Society of America (IDSA) clinical practice guidelines for the treatment of MRSA infections, which mention oral combination therapy with ST plus RFP for certain indications (e.g., bone and joint infections). We also cite the 2021 UK MRSA treatment guidelines, which note that while RFP combinations were once considered (especially for biofilm-related infections), there is no clear evidence supporting their efficacy in that context. Furthermore, as noted by the Editor, we have included the European Pediatric Bone and Joint Infection Guidelines which recommend ST plus RFP in specific scenarios. By incorporating these sources, we have ensured that our discussion is internationally relevant and provides a more comprehensive perspective beyond the Japanese context.

Major Point 4: Strain Characterization (MIC Discrepancies)

Comment (Editor): “Clarify the discrepancies in MIC values for your clinical strains, particularly regarding the strain with an intermediate MIC to TMP/SMX. Discuss the implications of these differences on your results and conclusions.”

Response: We investigated the noted discrepancies in minimum inhibitory concentration (MIC) values by performing additional experiments for clarification. We reconfirmed the MIC of ST for each clinical isolate using both the standard microdilution method and the E-test, alongside the quality control strain (ATCC 29213) to ensure accuracy of the methods.

For strain KAM636, which originally appeared to have an intermediate MIC of ST, the E-test revealed slight bacterial growth within the inhibition zone, corresponding to an MIC of 16 µg/mL. This finding explains the previous discrepancy, as 16 µg/mL indicates a resistant level (above the susceptibility breakpoint). We have corrected the reported MIC of KAM636 in the revised manuscript to 16 µg/mL and clarified that this strain is ST-resistant.

For the newly added clinical isolate KAM1160 (ST-low-susceptible strain), the E-test yielded an MIC of 3 µg/mL. According to the E-test manufacturer’s instructions, when an MIC falls between two standard dilution values, it should be rounded up to the higher two-fold concentration. Therefore, 3 µg/mL is interpreted as 4 µg/mL in broth microdilution equivalence, which falls into the resistant category (non-susceptible). In the revised manuscript, we now describe KAM1160 as a “ST low-susceptible (ST-LS) strain,” reflecting its borderline resistance.

We have updated the Results and Discussion sections to explain these MIC differences and their implications. Importantly, both clinical strains tested are now clearly identified as not fully susceptible to ST (one intermediate/low-susceptible and one resistant), and we discuss how the combination therapy efficacy was observed particularly in these non-susceptible strains. These clarifications ensure that our conclusions appropriately consider the variability in MIC results.

Major Point 5: Resistance Development

Comment (Editor): “Address whether you checked for the development of rifampin resistance in your experiments, particularly for the KAM444 strain that regained growth.”

Response: We have addressed this important point by analyzing whether rifampin resistance emerged during our chemostat model experiments. For each clinical isolate, we measured the MICs of ST and RFP for bacteria surviving after the 24-hour chemostat experiment. We found that in all cases the MIC of RFP increased substantially after exposure to rifampin alone, indicating that RFP resistance was indeed induced during the experiment (RFP monotherapy led to resistant subpopulations). We have added these findings to the Results and have expanded the Discussion to note that the development of RFP resistance is a significant concern when RFP is used alone, which further supports the rationale for combination therapy in our study.

Major Point 6: Clinical Relevance

Comment (Editor): “Provide a more nuanced discussion of the clinical applicability of your findings, particularly regarding the use of this combination therapy for TMP/SMX-resistant strains and in cases of sepsis or bacteremia.”

Response: We appreciate this suggestion and have revised the Discussion to more carefully frame the clinical relevance of our findings. First, we emphasize that our study is based solely on in vitro experiments, and thus we do not intend to directly recommend changes to current clinical treatment practices at this stage.

Second, we have added a nuanced perspective on potential clinical applicability. Specifically, we discuss that the ST plus RFP combination might be considered as a future therapeutic option in certain limited scenarios – for example, in situations where an MRSA infection is suspected but the organism’s ST susceptibility is unknown (and obtaining rapid MIC results is not feasible), and an effective oral regimen is needed (such as for step-down therapy or when IV treatment is not possible). In summary, the revised discussion highlights the potential value of our findings as a basis for future research; however, further data accumulation will be necessary before these findings can be applied clinically.

Minor Point 1: Figure Quality

Comment (Editor): “Improve the quality of Figure 1, which was reported as blurry.”

Response: We apologize for the suboptimal quality of Figure 1 in the initial submission. To resolve this issue, we have replaced Figure 1 with a high-resolution version in the revised manuscript.

Minor Point 2: Language and Formatting

Comment (Editor): “Address the specific language and formatting suggestions made by the reviewers, including: (a) Removing the word ‘therapy’ where inappropriate, (b) Correcting alignment issues in Table 2, (c) Reconsidering certain assertive statements, (d) Using passive forms where appropriate, (e) Properly referencing laboratory products.”

Response: We have made all the recommended language and formatting corrections in the revised manuscript:

• a. Use of “therapy”: We removed the word “therapy” in contexts where it was inappropriate for an in vitro study (specifically, we deleted “therapy” on original manuscript lines 44 and 312, where its use implied clinical treatment).

• b. Table 2 alignment: We corrected the alignment problems in Table 2. All columns and entries in Table 2 are now properly aligned and formatted consistently.

• c. Assertive statements: We revised an overly assertive sentence in the Discussion (original lines 273–274). The wording has been toned down to a more cautious statement, reflecting the fact that our conclusions are based on in vitro results with a limited number of strains.

• d. Passive voice: We have adjusted the manuscript text to use passive voice where appropriate to improve the formal tone.

• e. Referencing laboratory products: We have added proper references or source information for all laboratory products and reagents mentioned in the Methods.

We believe these changes have improved the clarity and readability of the manuscript in line with the reviewers’ suggestions.

Additional Suggestion 1: Number of Replicates

Comment (Editor): “Explain why only two replicates were conducted instead of three.”

Response: We appreciate the reviewer’s comment regarding experimental replicates. All experiments initially demonstrated high reproducibility with two replicates, supporting the reliability of our data. To enhance robustness, the revised manuscript includes a third replicate for the MIC assay and checkerboard assay, which confirmed consistency. For the chemostat model, due to the demanding nature of the 24-hour protocol, two replicates were conducted, both yielding highly similar results. These details are clearly described in the revised manuscript.

Additional Suggestion 2: Removal of a Sentence

Comment (Editor): “Consider removing or revising the sentence about a previous report (lines 260-261) as suggested by Reviewer 1.”

Response: We have removed the sentence in the Discussion referring to a “previous report” (along with the associated reference 19) as suggested.

Additional Suggestion 3: Referencing Web Address

Comment (Editor): “Add the web address mentioned on line 267 to your reference list.”

Response: In accordance with this suggestion (and Reviewer 1’s comment), we have added the previously unreferenced web address (from line 267) as a formal citation in the reference list (Ref. 23).

Response to Reviewer #1

Reviewer #1: “The study is generally well-planned and the manuscript is generally well-written. I have few recommendations about the manuscript.”

Response: We thank Reviewer #1 for the positive feedback on our study design and writing. We appreciate the specific recommendations and have addressed each point in the revised manuscript, as detailed below:

Comment 1 (Lines 44 and 312): “Remove the word ‘therapy’ because this is not a clinical study.”

Response: We have removed the word “therapy” from the relevant sentences in the manuscript (original lines 44 and 312), as suggested. We agree that since our work is an in vitro study and not a clinical trial, using the term “therapy” in those contexts was inappropriate.

Comment 2 (Line 47): “Write ‘against’ instead of ‘therapy of’.”

Response: We have revised the phrasing at line 47 in the manuscript. The phrase “the therapy of” has been replaced with “against” so that the sentence now correctly reads with “against” (e.g., “activity against MRSA”) rather than implying “therapy.”

Comment 3 (Table 2): “Some alignment problems exist in some columns of Table 2. These should be corrected.”

Response: We have corrected the alignment issues in Table 2 of the manuscript. All columns in Table 2 are now properly aligned.

Comment 4 (Lines 260-261): “I recommend the authors remove the sentence ‘In a previous report, RFP…’ and reference 19. In the cited report, although rifampicin and minocycline considerably decreased both WT and SCVs, both bacterial counts recovered to the initial number 48 hours later.”

Response: We agree with this recommendation. We have removed the sentence in question (about the previous report on rifampicin and minocycline) as well as the corresponding reference 19 from the Discussion. As the reviewer noted, that cited study’s findings (rifampicin + minocycline reducing MRSA counts only temporarily) did not directly support our discussion, and omitting it has made our narrative more concise and clear.

Comment 5 (Line 267): “A number can be given for the web address and this reference can be added to the references list.”

Response: We have added a numbered citation for the web address mentioned in line 267 of the original manuscript. Specifically, the web resource is now included as an entry in the reference list, and we cite it in the text by its reference number.

Comment 6 (Lines 273-274): “This sentence is a bit assertive. This was an in vitro study and the number of strains tested in this study was low. You can use a more appropriate sentence or you should support this sentence with information obtained from clinical studies.”

Response: We appreciate this comment and have revised the sentence in lines 273–274 to tone down the assertiveness. In the original manuscript, that sentence overstated the implications of our findings. We have rewritten it to be more cautious and context-appropriate, given that our evidence comes from an in vitro study with a small sample of strains.

Response to Reviewer #2

Reviewer #2: [General comment] “The article presents some intriguing findings, yet it is imperative to substantiate these outcomes with a more extensive sample size, encompassing both clinical and reference strains.”

Response: We thank Reviewer #2 for the overall positive view of our findings and the important suggestions. We have implemented changes to address the concerns raised. Below are our point-by-point responses to each comment:

Comment 1: Sample size of strains

“The article presents some intriguing findings, yet it is imperative to substantiate these outcomes with a more extensive sample size, encompassing both clinical and reference strains.”

Response: We completely agree that a larger sample size would strengthen the study. In response, we have added an additional MRSA clinical isolate to our experiments. Specifically, we included one more ST-low-susceptible MRSA clinical strain (KAM1160) and repeated the key experiments for combination efficacy with this strain. We have incorporated these new results into the revised manuscript. We have also explicitly acknowledged in the Discussion that the total number of strains tested is still small, and we have listed the limited sample size as a limitation of our study.

Comment 2: “In the event of a resubmission, other factors will have to be taken into consideration.”

Response: We understand the reviewer’s remark and agree that there are additional factors which could be important to explore in subsequent studies (for example, different strain backgrounds, resistance mechanisms, or in vivo conditions). For the current revision, we focused on the specific major points raised by all reviewers.

Comment 3: “It is preferable to utilize passive forms.”

Response: We have reviewed the manuscript for tone and voice, and we have revised many sentences to use the passive voice where appropriate.

Comment 4: “Please clarify why only two replicates were conducted, rather than three.”

Response: We thank the reviewer for this important comment. We have clarified this point in the revised Methods and Results. Most experiments initially showed high reproducibility with two replicates. To ensure robustness, a third replicate was added for the MIC and checkerboard assays, confirming consistent results. For the chemostat experiments, only two replicates were conducted due to the complexity and labor-intensive nature of the 24-hour protocol. Both runs showed nearly identical outcomes, supporting the reliability of the data.

Comment 5: “The sources of the laboratory products in question are not adequately referenced.”

Response: We have updated the manuscript to include proper source information

---

## [Editor Report · Decision Letter 1]

17 Apr 2025

Sulfamethoxazole-trimethoprim plus rifampicin combination therapy for methicillin-resistant Staphylococcus aureus infection: An in vitro study

PONE-D-24-38956R1

Dear Dr. Masaki Nakamura,

We’re pleased to inform you that your manuscript has been judged scientifically suitable for publication and will be formally accepted for publication once it meets all outstanding technical requirements.

Kind regards,

Abdelwahab Omri, Pharm B, Ph.D, Laurentian University

Academic Editor

PLOS ONE

---

## [Editor Report · Acceptance letter]

PONE-D-24-38956R1

PLOS ONE

Dear Dr. Nakamura,

I'm pleased to inform you that your manuscript has been deemed suitable for publication in PLOS ONE. Congratulations! Your manuscript is now being handed over to our production team.

Kind regards,

on behalf of

Dr. Abdelwahab Omri

Academic Editor

PLOS ONE